# Satellite data indicates recent Arctic peatland expansion with warming
K. A. Crichton [1] ✉, K. Anderson [1], R. E. Fewster [2], D. J. Charman [1], M. Garneau [3], M. Väliranta [4], M. Mleczko [1], J. N. Handley[1], S. Hodson[1], R. E. Parker [1], G. T. Swindles[2,5], M. Blaauw [2] & A. V. Gallego-Sala [1]

Northern peatlands are an important carbon store in mid to high latitudes, but become increasingly discontinuous in the higher latitudes, associated with temperature and precipitation limits on plant growth. During the last four decades, mean annual temperatures in the Arctic have increased on average by ~3 °C. Warmer temperatures and longer growing seasons likely drive increases in plant productivity throughout northern latitudes, but it is not clear whether warming has resulted in lateral spread of Arctic peatlands. Using long time-series Landsat satellite data, coupled with information gathered from fieldwork in situ, we show that Arctic peatlands have likely undergone lateral expansion over the last 40 years. On 21 transects from the edges of 16 extant peatlands in the European and Canadian Arctic (both high and low Arctic locations from 62 to 79°N), over two thirds of the peatland edges we studied showed statistically significant peak-summer greening (as Normalised Difference Vegetation Index) in the last 15 to 20 years, compared to the period 1985–1995. Peak summer moisture (as normalised Difference Moisture Index) levels remained stable or increased at most study sites. The lateral expansion of Arctic peatlands suggests they are an increasingly important natural carbon sink, at least in the near term.

Northern peatlands are an important present-day soil carbon store, containing ~415 ± 150 PgC (where 185 ± 70 PgC are found in permafrost peatlands), and occupying more than 3.5 million square km[1]. These high soil organic carbon stocks are a result of peat-forming plant communities that create carbon-rich soils due to low decomposition rates in often low temperature, waterlogged, anoxic conditions in the soil[2]. The size of this high-latitude carbon store, accumulating since the end of the last glacial maximum[3], equates to about half the amount of carbon in all global forest systems[4].

Arctic temperatures have increased almost four times faster than the global average over the last several decades[5], due to Arctic amplification, with mean temperatures rising by 3.8 °C between 1979 and 2021. Over the same period, satellite data have shown a general spectral "greening" trend (i.e., increases in satellite-derived greenness indices) in the Arctic region[6–8]; with far fewer areas showing spectral "browning" (reductions in satellite greenness indices). Recent warming has directly led to a reduction in (cold) temperature limitations to plant growth, and/or a growing degree day increase, and this has been linked to greening trends in high latitudes[9]. Shifts in vegetation communities have also been witnessed; for example "shrubi-fication" has occurred in recent decades in some high-latitude peatlands[10–14].

As well as temperature limitations on productivity, Arctic plant communities are sensitive to other climate, environmental and ecological condition changes, for example, permafrost thaw and shifts in hydrological conditions[15–17] as well as non-growing-season changes to soil nutrients and freeze-thaw cycles[17–19]. Soil ecosystem dynamics are affected both through the direct and indirect effects of warming, including increased primary productivity driving carbon input; but also increased soil carbon decom-position rates arising from permafrost thaw (and increased oxygen avail-ability linked to water table depth change) or increased bacterial respiration rates directly in response to warming[20].

Studies of past changes in peatlands have shown an increased carbon sink (i.e., increased soil carbon accumulation) in response to warming in northern high latitudes over the last ~10,000 years[21,22] and over the last ~1000 years[23,24]. More recent warming has resulted in variability in carbon accumulation in permafrost peatlands linked to recent ecohydrological dynamics[13,25–27]. Mod-elling studies suggest that Arctic peatlands will remain a carbon sink in the near-term in response to anthropogenic warming[28–30], but that Arctic peat-lands may become a carbon source from mid-century[30,31] linked to moisture changes (drying), permafrost thaw, and associated vegetation community

[1]University of Exeter, Exeter, UK. [2]Geography, School of Natural and Built Environment & [14]Chrono Centre for Climate, the Environment and Chronology, Queen's University Belfast, Belfast, UK. [3]Université du Québec à Montréal, Montréal, QC, Canada. [4]University of Helsinki, Helsinki, Finland. [5]Ottawa-Carleton Geoscience Centre and Department of Earth Sciences, Carleton University, Ottawa, ON, Canada. ✉e-mail: k.a.crichton@exeter.ac.uk

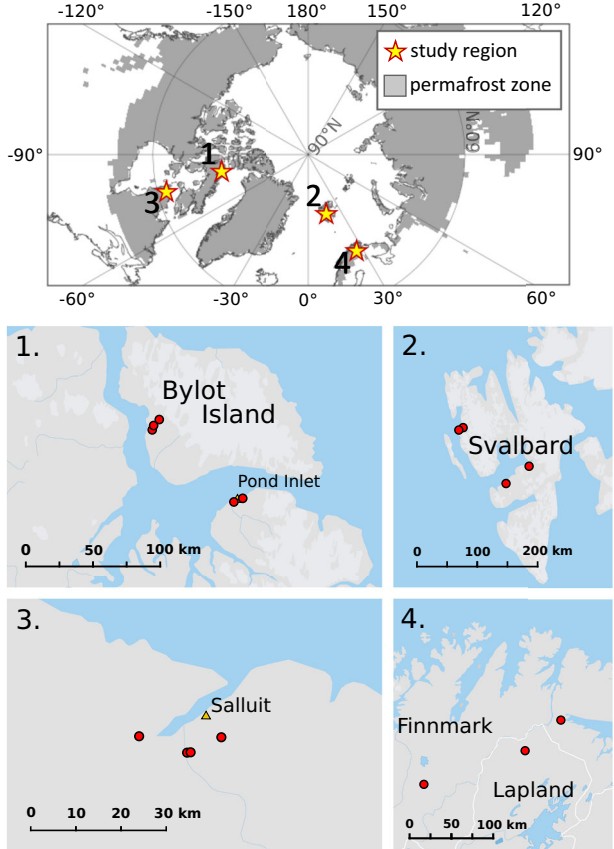

**Fig. 1 | Location of study regions (stars) and peatland sites (red dots).** Regions 1 and 2 are in the High Arctic and regions 3 and 4 in the low Arctic. Permafrost zone is from NCSCDv2 (the Northern Circumpolar Soil Carbon Database version 2, Hugelius et al.[46]). Maps created using Natural Earth data in QGIS. Site names are provide in SI A.

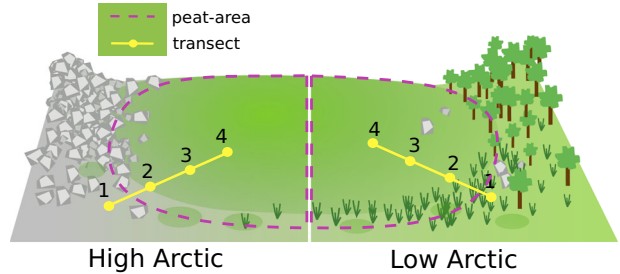

**Fig. 2 | Sketch showing peat-area and transects used in this study.** Example of a High Arctic site (left) where edges of peatlands are often (but not always) bare ground. Example of low Arctic site (right) where edges of peatlands are often (but not always) colonised by upland vegetation (herbaceous shrubs, small trees (only in Lapland)).

changes. These studies also suggest that only a strong mitigation scenario will maintain northern peatlands as climate neutral[30]. The vegetation composition of Arctic peatlands is spatially heterogeneous, in large part due to permafrost dynamics and the knock-on effects on peatland water budgets. Due to the complexity (driven by spatial scales of heterogeneity) of Arctic peatland eco-systems, numerical models are, of necessity, simplifications based only on best (and robust) current knowledge, and they do not currently well-integrate the retroactive feedbacks from the different changes. In effect, the representation of dynamic peatlands in land surface models is still in its infancy, and data available for validation of these models are sparse. As such, future predictions of the response of high latitude and Arctic peatlands to climate drivers are still highly uncertain[1,2,19,32–34].

We previously found that Arctic peatland sites have mostly "greened" (increased plant productivity) over the period 1985 to 2020[19], with strongest greening found in the high Arctic locations, linked to improved growing conditions. In this study, we test whether recent lateral expansion in a series of Arctic peatlands can be detected using satellite data.

Here, we consider changes over the last 40 years on the peatlands and along the edges of peatlands in both high Arctic and low Arctic regions spanning different continents covering a range of Arctic and sub-Arctic zones, totalling 16 individual sites in which 21 transects were sampled (Fig. 1). Satellite data from the Landsat TM and ETM + sensors were used, with field validation from sites from the NERC project "Increasing Carbon Accumulation in Arctic Peatlands (ICAAP)" to test whether the peatlands have expanded laterally. To do this, we considered surface reflectance-data metrics from the Landsat Collection 2 Archive for productivity (normalised difference vegetation index (NDVI)) and moisture content (normalised difference moisture index (NDMI)) in peak-summer (see methods and SI

B). We quantified changes on the sampled peatlands, comparing the period between 1985–1995 for all sites to the period 2005–2015 for the high Arctic, and 2010–2020 for the low Arctic (the different recent periods are due to data quality and availability, see "Methods" and SI B). We considered two characteristics of the peatland sites: 1. Changes seen on the peatland general area notated as "peat-areas"; and 2. Changes seen along edge-to-peat transects. Changes in productivity and moisture on the peat-area provides a context and broader picture with which to compare changes identified along the edge-to-peat transects (Fig. 2). Field sites, peat-areas and transects are fully described in SI A. We limit the number of study transects to those identified in the field site-visits, to eliminate a source of great uncertainty in attempting to create additional transects based only on our remotely sensed data. In this method, we are not equating surface changes with below-ground carbon accumulation, but we are considering the dynamics of peatland forming plant communities, that can give an indication of possible lateral (areal) changes of the peatland.

## Results

All peat-areas show increases in mean NDVI between the early (1985–1995) and late period (2005–2015 for high Arctic, 2010–2020 for low Arctic) (Fig. 3B), despite the differences in plant communities and other conditions at our sites (see SI A). Greatest increases in NDVI are found in the Svalbard locations, which also show the greatest increase in mean temperature between the two periods (Fig. 3A). The peat-areas with higher increases in NDVI also generally show higher increases in NDMI (moisture) (Fig. 3B). High Arctic peat-areas show increases in peak-summer moisture (NDMI), while three out of seven low Arctic peat-areas show decreases (Fig. 3B). The distribution of the changes seen per pixel in each region and peat-area (Fig. 3C) show a general shift from lower to higher NDVI for peat-area pixels in all regions (shown as shaded areas in the lower left quadrant—a reduced % of lower-than-mean NDVI pixels; and shading in the upper right quadrants—an increased % of higher-than-mean NDVI pixels. Full NDVI and NDMI distributions are shown in Supplementary Fig. S8). For NDMI, the picture is more mixed, with some pixels within the peat-areas showing decreasing NDMI compared to the 1985–1995 mean (shown as shading in the lower right quadrant and upper left quadrant). This trend is slightly more evident in the low-Arctic sites, especially Lapland.

The per-site transect results for the early and late periods for NDVI and NDMI (Fig. 4) show that for both the early and late periods, edges are drier than locations further into the peatland; the transects are characterised by a reduction in peak-season NDMI in all locations moving towards the peat edge (with the except of Kevo 1). This is also generally the case for NDVI, with NDVI decreasing towards the edge, but it also depends somewhat on vegetation cover; in areas where mosses transition to grasses/shrubs (e.g., Lapland, see SI A), the edge NDVI does not vary greatly from the in-peat pixel value. This can be seen in the difference between Lapland edge and on-peat pixels values (small), and the difference between Svalbard edge and on-peat pixels values (larger).

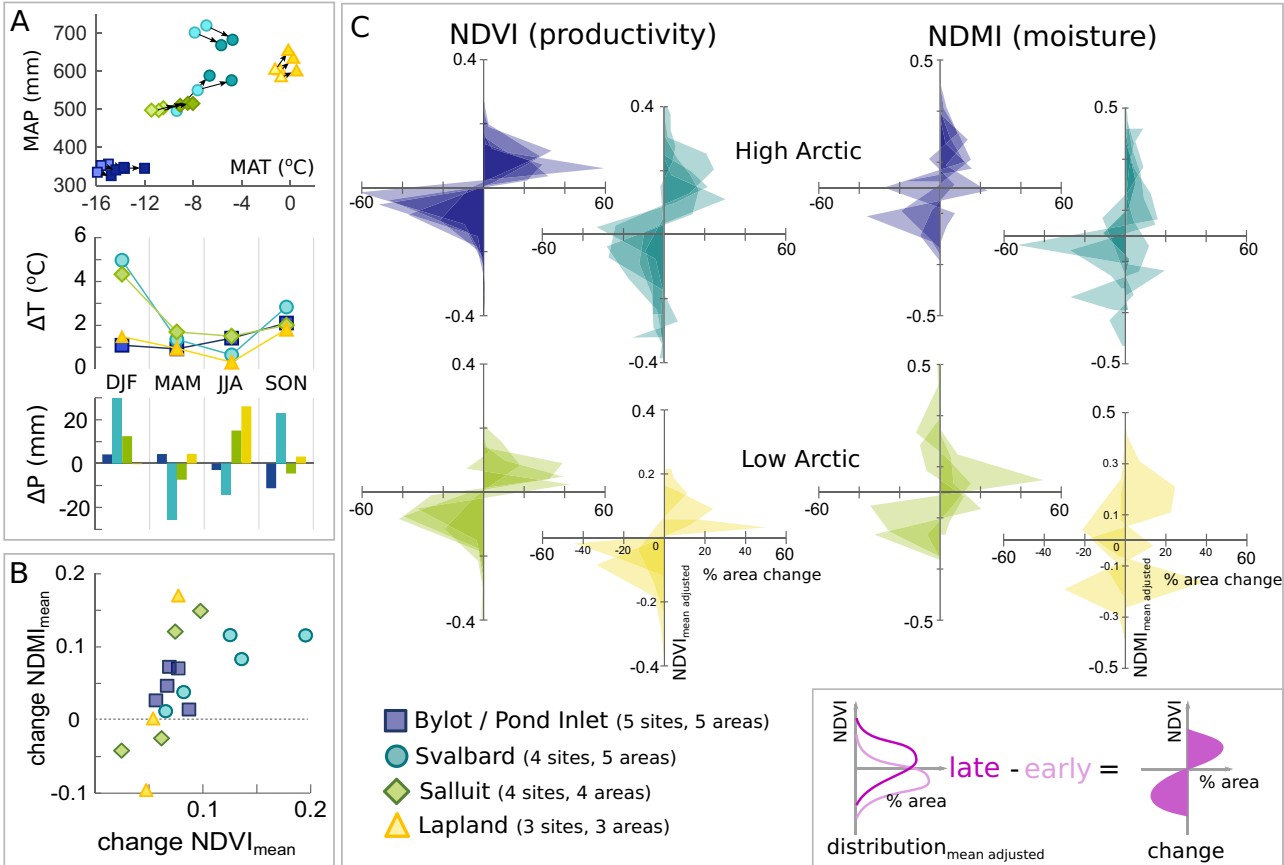

**Fig. 3 | Changes in peak summer NDVI and NDMI in peat areas between the early (1985–1995) and late periods (2005–2015 for High Arctic, 2010–2020 for Low Arctic), coloured by region. A** Temperature and Precipitation for the study sites from ERA5 reanalysis data (Hersbach et al.[63]), showing mean annual values (Mean Annual Temperature, Mean Annual Precipitation) for the early (lighter shaded) and late (darker shaded) periods, and seasonal mean change between early and late periods for the region means (grouped in 3-months periods). **B** change in mean NDMI (moisture) and mean NDVI (productivity) between early and late periods. **C** changes in areal distribution of NDMI and NDVI in peat-areas, with inset graphic to aid interpretation of the figure.

Considering the changes between the early and late period; out of 21 transects, 14 (67%) edge pixels (points 1 in Fig. 4) show statistically significantly higher NDVI in the late versus the early period, compared to 11 (52%) of the furthest-in-peat pixels (points 4 in Fig. 4) (with 12 (57%) for point 2, 11 (52%) for point 3). No points showed statistically significant reduction in NDVI. Out of 21 transects, five (24%) edge pixels show statistically significantly higher NDMI (point 1), seven (33%) for furthest-in-peat (point 4), (6 (29%) for point 2, 8 (38%) for point 3). Only two points in any position on the transects showed a statistically significant reduction in NDMI (point 2 Pond Inlet East, and point 4 Blomstand 1).

Variability in NDVI and NDMI between sites is generally higher in the high Arctic locations (Fig. 5A, showing the region mean values), mirroring the field site observations on site heterogeneity (see SI and discussion). The absolute values of NDMI appear to depend on the local site conditions for the in-peat pixels (points 2 to 4, Fig. 5a). Increases in both NDVI and NDMI between the early (1985–1995) period and late period are generally higher in the High Arctic regions than the Low Arctic regions (Fig. 5b). Increases in both NDVI and NDMI are similar for edge and in-peat pixels on a per-region basis, indicating a fairly uniform increase in peatland plant productivity along the transects, including at the edges.

## Discussion
### Representativeness of the study sites
Our study encompasses a wide range of pan-Arctic peatland types across the low and high Arctic of Europe and North America, with permafrost extents in our sites ranging from sporadic (e.g., Suossjavri), to discontinuous (e.g., Karlebotn, Kevo) and continuous (e.g., Colesdalen, Peat Qilaliariak). Our

permafrost-containing sites have typical permafrost peatland landforms, including high and low-centred ice-wedge polygons (e.g., Bylot Island and Colesdalen 1) and palsas/peat plateaus (e.g., Karlebotn), but are increasingly becoming complexes with thermokarst and unfrozen areas that are reminiscent of fens and bogs. Our study considers 16 sites and 21 transects, set out on two larger low-to-high Arctic transects (Europe, and Canada) but there is now a pressing need to extend this type of analysis to new sites with the benefit of a-priori knowledge of true peat-edges. The Landsat archive represents an excellent resource for this analysis, as we demonstrate here.

Evident from both field validation and satellite data, is the heterogeneity of the different Arctic regions we considered, especially between high and low Arctic, but also the heterogeneity within a region itself. For example, in the Canadian High Arctic region, our study areas located less than 100 km apart in Bylot island and Pond Inlet are characterised by very different landscapes and peatland characteristics. In our high Arctic sites, especially Svalbard but also Bylot/Pond Inlet, we observed mainly low-lying, coastal peatlands, with peat forming over bedrock, rocks or glacial silts in gently sloping ground towards the coastline, surrounded by tundra vegetation of varying density. In Salluit, we also analysed shallow coastal peatlands, which have accumulated on sands or rocks and with sparse surrounding vegetation. Salluit, although located at a much lower latitude, has an annual mean temperature that is more similar to Svalbard than Lapland (Fig. 3A), but with higher summer temperatures and no periods of 24 h daylight (midnight sun)[19]. Likely due to these cold conditions, vegetation was sparse compared to the warmer sites in Lapland. In Lapland, we measured discontinuous permafrost peatlands, including palsa mire complexes, which were often surrounded by higher ground with tree cover.

**Fig. 4 | Landsat TM and ETM sensor data for each transect showing early period (1985–1995; left boxes) and late period (right boxes; 2005–2015 for High Arctic, 2010–2020 for Low Arctic) peak-summer values, as box and whisker plots.** Boxes shaded-in represent statistically significant changes (>95% confidence) for greenness (for NDVI) or moisture (for NDMI). Site names shown in bold over each subplot, more information on each site is give in SI A.

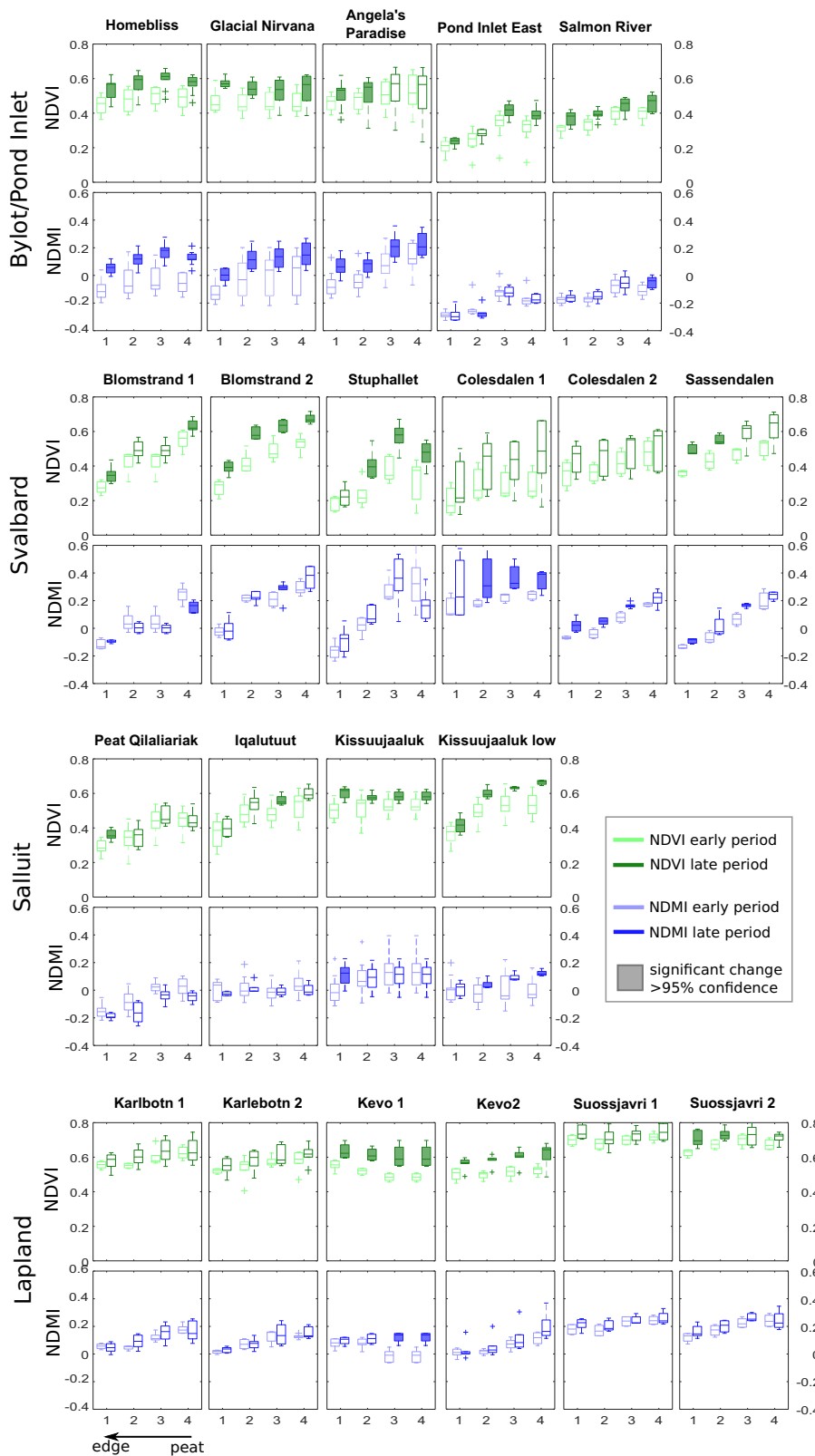

These Lapland peatlands are more extensive, with generally deeper peat, and they also grow over glacial material or directly on rocks. We observed quite large variability between sites in each region, and except possibly for Lapland, these inter-regional differences were often greater than the intra-regional differences between sites. More details on site characteristics are provided in our supplementary material (SI A). Despite these site-scale differences in vegetation and climate, we observe similar direction of change across regions, with increased greenness (NDVI) found across the transects, and a general stability in peak-summer moisture (NDMI).

## How measured NDVI and NDMI can indicate changes in peatland extent

From the general increase in NDVI seen on the peat areas, combined with increased NDVI seen at peat edges, and a maintenance of peak-summer

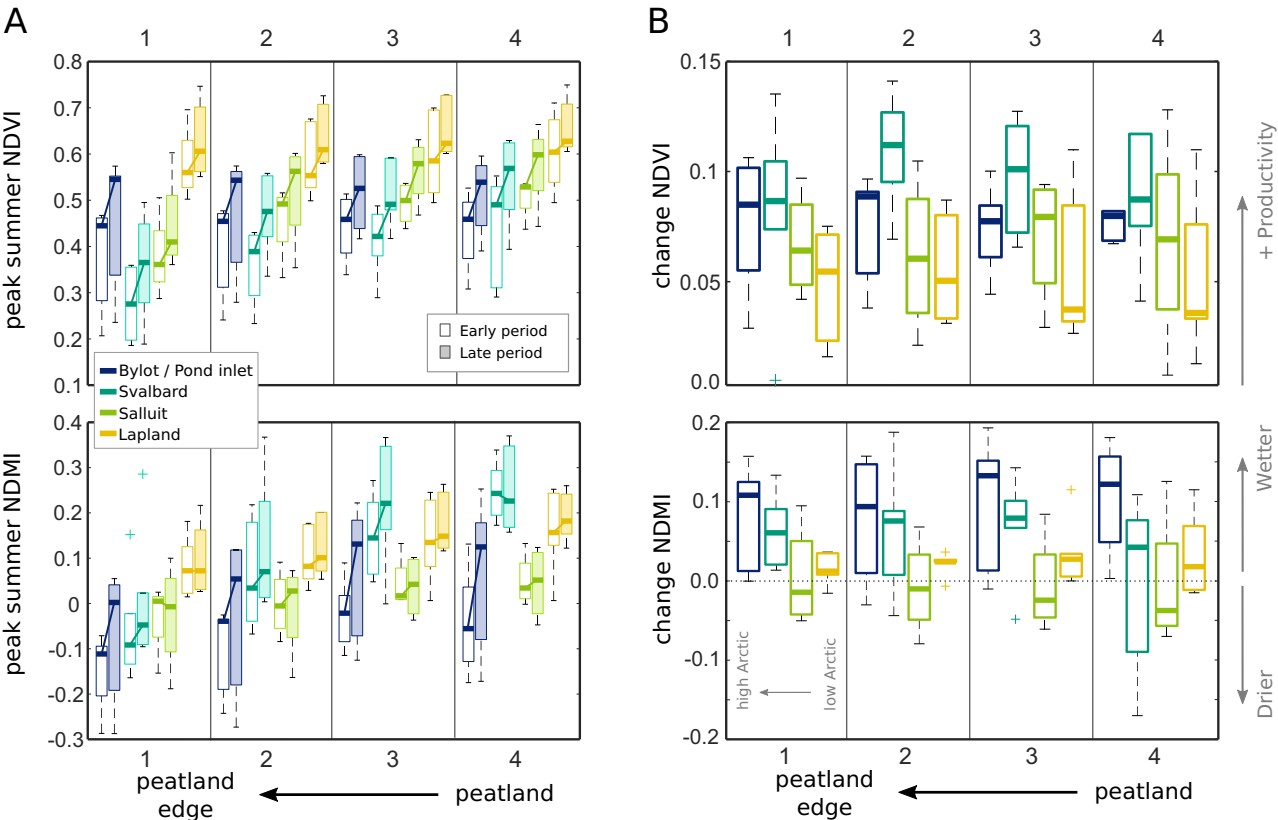

**Fig. 5 | Moisture (NDMI) and productivity (NDVI) indicators along the transects, grouped by region.** Box plots showing median value and range of peak summer NDVI and NDMI for peat-to-edge transects, Site-mean values grouped by region (**A**) and change in the mean NDVI and NDMI (**B**) between the early (1985 to 1995) and late period (2005 to 2015 for High Arctic; 2010 to 2020 for Low Arctic), summarising Fig. 4. A larger distribution indicates higher variability between sites in the region.

NDMI, we conclude that during the early period (1985–1995), the edge pixels were less vegetated than they were more recently. At the date of our site visits, the peat edges (or other points on our transects) were not waterlogged, but it is possible that in the past, some of the area was waterlogged and contained ponds and pools (likely for some Bylot Island, in-peat pixels). If pools had since been terrestrialised and infilled, this would register as an increase in NDVI. However, visual inspection of the Landsat images and a general increase or stability in NDMI (as our data indicates) since the early period (i.e., higher or stable moisture) do not suggest to us that the increase in NDVI seen generally is due to in the infilling of pools since the earlier 1985–1995 period (i.e., the sites are not dominated by pools in the 1985–1995 period). Assuming then that pool and pond infilling is not the driver of NDVI increase, possible explanations of increased NDVI over the Landsat edge pixels may fall in to four cases: 1, a shift from lower to higher productivity plants; 2, greater leaf area (increased productivity), but fixed plant density; 3, greater number of plants (increased density), but productivity per plant remains fixed; 4, a shift from very few high productivity plants to many low productivity plants; or a combination of these factors. Shrubification is an example of case 1, a shift to higher productivity plants (as shrubs often have higher NDVI than peat-forming plants, mosses[35]). From our site visits and drone data, we know the peat edges at our sites are not presently dominated by shrubs, especially in the high Arctic where shrubs are completely absent (see SI A), so we discount shrubification as an explanation for the measured NDVI increase. All the other cases are aligned with the conclusion of expansion of the peatland forming plants, which we interpret as an expansion of the peatland.

The spatial grain of Landsat data (30 × 30 m per pixel) deserves consideration because each pixel will consist of a range of plant types and, especially at peat edges, of a range of land cover types. In the high Arctic, less vegetated locations (such as Blomstrand and Stuphallet, and Pond Inlet) edge pixels will likely comprise a mix of sparse vegetation and bare ground.

In Low Arctic or more vegetated sites (Lapland especially), edge pixels will likely comprise a mix of peat-forming plants, grasses, with some shrubs or possibly trees. These characteristics are indicated in the transects' absolute NDVI values' variability, as discussed in the results. Despite this heterogeneity between our sites, we still see a similar pattern: peat-area greening, across-transect greening, and a general maintenance of peak-summer moisture.

As a secondary line of evidence to support our conclusion of peatland expansion, we have considered preliminary ¹⁴C data from a number of peat cores that were collected on our transects as part of an upcoming study (in Prep). The methods for peat core collection and radiocarbon calibration are provided and described in Supplementary Material C. Each of these cores have calibrated mean ¹⁴C peat-basal dates that fall between the start of the Landsat dataset, 1985, and the present (Supplementary Fig. S10). This chronological data shows that across our transects, peat, and thus soil organic carbon, has very likely been actively accumulating since 1985, with recent accumulation evident both within the peatland and at the edges (Supplementary Fig. S9). Dating material using ¹⁴C results in an estimation of age, and we report the range of the calibrated dates in Supplementary Table S4. However, taken together with the satellite data showing surface greening during the same period, these two lines of evidence provide corroboration that a likely explanation is the expansion of the peatland.

**Changes observed per region**

By region, the greatest increases in NDVI along the transects are those that have seen the highest increases in summer temperature (Svalbard, and Bylot Island/Pond Inlet) (Figs. 3 and 5). As example, the region with the greatest increases in mean NDVI since the 1985–1995 period is Svalbard (Fig. 5), this region has also seen the greatest (absolute) increases in summer (and winter) temperature (Fig. 3). Svalbard peatlands do not show the highest edge-pixel absolute NDVI values, likely due to the peatland edges being a mix of

vegetation and bare ground (Fig. 5) but they do show the greatest increases all along the transects, including the edges. Indeed growing season temperature has been linked to productivity in previous studies[9,19].

The regions with highest increases in NDVI, also have the higher increases in NDMI (Fig. 3B), underlining that as a site becomes greener it also becomes wetter, noting that there is a relationship between the peat-area NDVI and NDMI changes, as both indices use reflectance in the NIR band, and both are linked to plant health. The sites with lower increases in NDVI (particularly Lapland) also showed lower increases in NDMI (Fig. 3B) between the early and late periods. For the peat edges, these patterns also hold fairly well (Fig. 5B), but do appear to depend on baseline NDMI levels – for example, Svalbard sites on average show greatest increases in NDVI, but not NDMI, but they started with the highest peak-summer NDMI levels. A factor likely affecting NDMI will be sub-surface conditions, such as permafrost thaw, which is especially important in palsa mires, such as the Karlebotn Lapland site. A recent data compilation[36], including aerial photo survey, showed that 60–70 % of palsas have thawed in Finland over the last several decades (since 1950 or so) resulting in wetter soils. Other parts of Lapland have also evidenced widespread palsa/peat plateau degradation in recent years[37–39], although the Karlebotn site currently appears relatively intact.

A complicating factor for the Lapland sites in general is the presence of discontinuous permafrost landscape features, including palsas; where thawing can affect both water availability and local topography, for example creating slumps and standing water bodies. Despite this, in our study there is still a recent increase in greenness (NDVI) along the full transect in the Lapland sites since the 1985–1995 period, and also on the total peatland-area (Figs. 3 and 4), but less strong than for other regions (Fig. 5). We cannot rule out that this may in part be due to possible encroachment of vascular plants or shrubs, as they are present at the sites. However, peatlands are persistent and resistant in this region, and recent evidence shows sustained lateral expansion over recent millennia and up to the present day[40]. The transects on these sites exhibit stable peak summer moisture levels from the earlier to the later period (Figs. 4 and 5). An increasing summer precipitation trend at this site[19] could explain the persistence of high soil moisture, along with possible permafrost thaw (which we did not measure in this study), supporting the dominance of peatland plant communities. Patterns of peak summer moisture levels (as NDMI) at any site will depend on precipitation, but also evaporation rates, riverine or stream input to the site (so, topography and location), including spring snowmelt conditions, and therefore also snowpack accumulated in the winter and spring.

## Possible future of Arctic peatlands

The warming of the high latitudes has, and will continue to, reduce temperature limitations on plant growth, potentially changing the shape of plant communities in these locations where currently, wetland and peatland plants dominate, with a potential to drive greening trends[9]. Much work has been done to understand the drivers of Arctic plant community dynamics, in order to improve understanding of future structural or functional change[19,41,42]. One strong point of agreement is that drying is detrimental to peatland and wetland plant communities. The presence and persistence of a shallow water table are essential to maintaining peatland communities' dominance over other plant types.

Future projections of Arctic hydrological conditions show increases in summer and autumn precipitation[43], but also increases in open water evaporation with warmer temperatures. The increased precipitation is likely to benefit peatlands and wetlands, especially ombrotrophic wetlands, and may help to maintain high latitude peatlands as overall carbon sinks, (again with the likely corollary that future warming is kept to a minimum[30]). Indeed, our dataset shows summer moisture maintained or increased over the recent past, and a greening across the whole peat area, including edges. Studies agree, however, that on the southern edge of the discontinuous permafrost zone (such as the Lapland sites), warming to date and even the lowest future emissions scenarios have already committed huge areas of the sub-arctic peatlands to now-unavoidable permafrost thaw[1,44,45,46]. This will likely have a huge effect on how low Arctic

landscapes function, affecting soil hydrological conditions, and nutrient availability, as well as changing landscape topography in some cases. The permafrost thaw may turn some peatland areas into carbon sources[47], but here again there is uncertainty. For example, a study of degraded (thawed) permafrost in northwestern Canada showed increased carbon uptake driven by a shift to *Sphagnum*-dominant vegetation responding to improving growing conditions[27].

In terms of the dominance of peat forming plants and wetlands over other vegetation types in future climate, the moisture regimes will be determinant[48]. As the future Arctic is likely to become wetter[43] with increased precipitation during the growing season and during autumn, a period identified as important for peatland greening[19], peatlands have a chance of persisting and thriving here. Further than this, carbon accumulation is linked to increased PAR0 (photosynthetically active radiation over the growing season, an indicator of how much useful radiation the plants get during the growing season) as evidenced in peat cores[23]. This carbon accumulation occurred regardless of whether these peatlands were moss dominated bogs or fens, and whether the dominant plant functional types changed over time, indicating an increased soil carbon sink even with plant community changes. Arctic peatlands are projected to become stronger carbon sinks until the end of the 23rd century, even discounting further expansion from present-day area[24]. Across the northern hemisphere, peatland initiation since the last glacial maximum has been driven by warming growing seasons, rather than any increase in effective precipitation in recently deglaciated locations (e.g., North America), but for Western Siberia peat initiation may have been driven by increased net precipitation at about 11k yr BP[49]. Furthermore, moisture was not found to have a control on global-scale variations in (vertical) peat accumulation, but acted rather as an "on-off" switch[24]. As the whole of the Arctic zone is warming, and will continue to warm in the near term even with emission reductions[50], together with a projected increased growing season precipitation[43], it thus seems possible that further expansion of existing peatlands may occur, in addition to new peat initiation at sites where soil moisture conditions become favourable.

In this study, however, we cannot make definitive statements on the impact of the observed increases to productivity and inferred peatland expansion and soil carbon accumulation, because we do not consider soil respiration or the drivers of moisture changes at each site. However, we do have evidence that indicates new peats, and thus soil organic carbon, have been actively accumulating at these sites since 1985, even at the edges (SI C). For the Arctic region as a whole, the question of whether peatland expansion translates into maintaining Arctic wetlands as a net carbon sink depends on the relative rates of change. New peat carbon accumulation at the edges (including of any individual peatland, or considering the relative balance between carbon gain and loss that may exist between the northern and southern limits of the Arctic peatland zone) needs to be outpacing any losses of existing/old peat carbon that could occur due to permafrost thaw[51,52], wildfire (that can cause rapid carbon loss[53,54], increased microbial soil decomposition resulting directly from warming[20,55,56] and land use change. More empirical data are needed to better understand the drivers of change in the Arctic terrestrial carbon cycle, as well as to map pan-Arctic permafrost features, and their rate of change, in order to make better predictions of the future Arctic-peatland carbon pool.

## Conclusion

Using Landsat data, we found that present-day Arctic peatland edges have been greening since the 1985–1995 period (as increasing NDVI), indicating a possible expansion of Arctic peatlands over the last 40 years, likely as a response to warming. At the same time, in most cases, peak summer moisture (indicated via NDMI) is stable or has also increased, likely supporting the persistence of peat-forming plant communities. This conclusion is supported by [14]C evidence of recent carbon accumulation below-ground at our study sites. This recent expansion of Arctic peatlands has implications for land surface models that do not include a dynamic peatland component capable of expansion or shrinkage, as these models are unlikely to capture the actual change in the overall peatland carbon balance.

As warming continues, we suggest that Arctic peatlands may continue to see increasing productivity and lateral expansion, at least in the near term, and that this process could be a key part of a global terrestrial negative feedback to anthropogenic climate change. Specifically, if expansion leads to increased carbon accumulation through the conversion of plant material into soil carbon, and its efficient storage due to low soil decomposition rates in high water table, anoxic conditions. However, from satellite data alone, we cannot assess soil carbon accumulation rates, so we cannot answer whether lateral expansion or increased productivity is resulting in long-term increases in soil carbon accumulation. Conversely, studies of soil carbon accumulation from peat cores cannot attribute whether measured increased soil carbon accumulation is due to reduced decomposition rates or due to increased carbon input via increased productivity. This study, finding recent increased productivity coinciding with recent warming, may provide part of the answer to that question.

## Method

We make use of multispectral satellite data dating back to the year 1985 to determine changes in plant productivity in Arctic peatlands. Due to difficulty in automatic classification of land cover types using remotely sensed satellite data alone, particularly in high latitude regions and at coarse spatial resolution[57–60], we made use of the NERC project (ICAAP) field site visits as validation to define the limits and edges of the peatland sites. The transects used here correspond, as best as possible, to those in the field where peat cores were extracted, with transect lengths lengthened to account for satellite data pixel size (30 m) (see SI B).

### Satellite data analysis

We used a readily available and widely used vegetation index (NDVI) as a proxy for vegetation productivity, or more specifically of photosynthetic activity, and the NDMI as an indicator of near-surface moisture. NDVI, in ratioing reflectance in the red and near infra-red (NIR) bands, is considered a suitable index to measure photosynthetic activity and has been used to do so in other high-latitude systems[60]. The NDMI metric represents moisture conditions in both soil and vegetation, as the index ratios NIR and short-wave infra-red reflectance. NDMI has previously been applied to study arctic systems moisture content[61], and we consider it a suitable surface moisture proxy for this work since the vegetation at these arctic sites has short stature and is regularly water-logged or high in moisture content being comprised of communities dominated by bryophyte species. For this reason, we consider NDMI to represent the general moisture conditions both in plants and soils for these communities. To measure these indices, we selected data points from the peak summer growing season only (selected based on inspection of changes in NDVI, as day 185–215 for Svalbard; day 190–210 for Lapland; day 190–220 for Bylot Island and Pond Inlet; day 200–230 for Salluit). For the productivity measure, this peak summer corresponds to peak NDVI values. For the moisture measure, the peak summer does not usually correspond to peak-NDMI values, as moisture is dependent on snowmelt, summer precipitation, and permafrost thaw. In any case, choosing peak summer for both indices allowed consistent tracking of ecological and hydrological dynamics.

In order to quantify changes in peatlands over time, we compared the early 1985–1995 period peak-summer mean NDVI and NDMI with a more recent 10-year period peak-summer-mean (2010–2020 for the Low-Arctic, 2005–2015 for the high Arctic) so as to separate the effect of natural variability from the effect of longer-term sustained changes. Ideally these early and late periods should be as far apart in time as possible, in order to be able to capture changes that may be relatively small. However, data availability in the form of sufficiently cloud-free conditions, and a sufficient number of images that passed quality checks, meant that for the High Arctic, the more recent period was set as 2005–2015, while with more data points available, the low Arctic recent period was chosen as 2010–2020.

The image set leveraged comprised scenes from Landsat 5 TM (L5) and Landsat 7 ETM + (L7) sensors, collection 2. Landsat Collection 2 is the state-of-the-art, newly reprocessed dataset, which has improved harmonisation geometrically and radiometrically, hence is best suited to long-time-series analyses[62]. Although there is also a newer Landsat 8 satellite with the Operational Land Imager sensor, the regions of the spectrum sampled do not match those for the TM and ETM +, making comparisons difficult, so we did not use the data (more details here[19]). A discrepancy was found[7] between L5 and L7 in measurements of NDVI over high latitudes of North America, resulting a very slightly higher NDVI calculated from L7 data compared to L5 for the same scene (on the order of 0.01 NDVI[19]). However, for consistency, we did not apply their correction to the L5 data for NDVI here, as we also calculated NDMI, for which we did not have a simple correction function.

Two general characteristics were examined to determine whether Arctic peatlands are expanding laterally. 1. Changes seen on the peatland overall are notated as "peat-areas", and 2. Changes seen along edge-to-peat transects. Changes in productivity and moisture on the peat-area provides a context and broader picture with which to compare changes identified along the edge-to-peat transects (Fig. 2). The peat-areas (1.) consisted of 200 to over 1000 landsat pixels (depending on the site), with pixel size of 30 m giving area sizes ranging from 180 to 900 km$^2$ (see Supplementary Figs. S4 to S7). We considered changes in the distribution of NDVI and NDMI between the early and recent periods, as well as the change in the mean values. The transects (2.) consisted of four points, running from the edge of the peat site towards the centre, where each point is one Landsat pixel (images of these transects are available in Supplementary Methods B, and Supplementary Fig. S9). We considered changes in the mean NDVI and NDMI between the early and recent periods. The location of the peat areas identified on the satellite image was selected during field campaigns and using drone-captured aerial photographic data having centimetric spatial resolution in 2019 for the European sites, and in 2022 for the Canadian sites (DJI Mavic Pro 2 in Europe; DJI Mavic Mini 2 in Canada). Peat edges were identified in the field, and transects were defined for peat monolith extraction for the ICAAP project. We use these same transects, but were obliged to extend them to account for the Landsat pixel size (30 m). The transects were extended in the direction of the peat where sensible. More details are available in SI B. As such, we are confident that in the present, these transects do represent the location of the peat to peat-edge.

### Data treatment

For the peat-areas, the distribution of NDVI and NDMI was calculated for the early and late periods. In Google Earth Engine, all Landsat 5 and 7 images within the peak-summer period were selected for the years of interest in the peat area of interest (an image-wide cloud pre-filter was also applied, and is described in full in SI B). Any pixels classified as cloud or cloud shadow were masked, and then the NDVI and NDMI values were calculated. Over the two periods, the maximum value per pixel was used to create the period-representative layer. The distribution values of the indices were found by binning in to 0.1 increments. To create the % change Fig. 3c, the early period (1985–1995) mean-adjusted (the mean calculated using both early and late period values) distribution was subtracted from the late period (2005–2015 high Arctic, or 2010–2020 low Arctic) mean-adjusted distribution, and converted into % change. For the transect points, the data were also obtained in Google Earth Engine, by defining the latitude and longitude of the four transect points and again extracting for summer-season data. These data were filtered for fidelity (including cloud cover), and annual mean values were calculated. A full description of this process is provided in SI B. The box plots shown in Fig. 4 are for all annual-mean data points within the peak-summer period for Landsat 5 and 7, split in to early and late periods. In Fig. 5A the data mean is grouped by region, and 5B shows the change in the mean value per site (so, late minus early period mean) and grouped by region. In these box plots, the central mark is the median value, the top and bottom of the boxes are 75 and 25th percentiles, respectively, and whiskers are the extreme values, not considered outliers. Outliers are plotted individually with a + symbol. These plots were created in MATLAB version R2023a, using the boxplot function. The statistical significance of the changes in transect NDVI and NDMI levels between the

early and late period (as summarised in Fig. 4) was performed using a Mann–Whitney *U* test.

## Reporting summary

Further information on research design is available in the Nature Portfolio Reporting Summary linked to this article.

## Data availability

All satellite data can be reproduced using the freely available Google Earth Engine platform and the described methods. Processed satellite data values are available in supplementary data. https://developers.google.com/earth-engine/datasets/catalog/LANDSAT_LT05_C02_T1_L2 https://developers.google.com/earth-engine/datasets/catalog/LANDSAT_LE07_C02_T1_L2.

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

## Acknowledgements
This work was supported by the UK Natural Environment Research Council under agreement number NE/S001166/1. We would like to thank those who granted us access to the Sirmilik National Park and the long-term ecological monitoring on Bylot Island, and the Mittimatalik and Salluit territories. This includes the Inuit communities of Mittimatalik and Salluit, and Park Canada staff. The Polar Continental Shelf Program (Natural Resources Canada) provided and organised logistical support for the fieldwork. This study was supported by the Natural Sciences and Engineering Research Council of Canada (Discovery program), the Canadian Arctic Network of Centers of Excellence, and the ArcticNet Northern Scientific Training Program (Polar Knowledge). We would also like to acknowledge the Centre for Northern Studies (CEN) at Laval University, and the Universities of Quebec in Trois-Rivières and Montreal for providing the researchers with funding and remote area safety expertise. Part of this research was conducted under Scientific Research Licence No. 02 045 22N-A, issued by the Nunavut Research Institute. We gratefully acknowledge the support and cooperation of the Nunavut Research Institute and the local Inuit communities. All research activities were carried out in accordance with the terms and conditions of the licence.

For the purpose of open access, the author has applied a Creative Commons Attribution (CC BY) licence to any Author Accepted Manuscript version arising from this submission.

## Author contributions
A.G.S., K.A., D.C. and K.C. conceived of the idea; K.C. designed and carried out the analysis; R.F., J.H., S.H., R.P. and M.M. provided material input to the study for peat core data and drone data; K.C. created the first draft of the manuscript; K.C., K.A., R.F. and A.G.S. were the core writing team; M.G., M.V., G.S. and M.B. are project partners and contributed to the manuscript. All authors provided critical feedback for the manuscript. A.G.S. was responsible for the whole project.

## Competing interests
The authors declare no competing interests.
