## [Transparent peer review file · Communications Earth & Environment]

Satellite data indicates recent Arctic peatland expansion with warming

Corresponding Author: Dr Katherine Crichton

Version 0:

Decision Letter:

Dear Dr Crichton,

Your manuscript titled "Satellite data indicates recent Arctic peatland expansion with warming" has now been seen by 2 reviewers, whose comments are appended below. You will see that they find your work of some potential interest. However, they have raised quite substantial concerns that must be addressed. In light of these comments, we cannot accept the manuscript for publication, but would be interested in considering a revised version that fully addresses these serious concerns.

We hope you will find the reviewers' comments useful as you decide how to proceed. Should additional work allow you to

- address these criticisms (that is, either to incorporate the suggestions or provide a compelling argument why the point made by the reviewer is not valid or relevant to the editorial threshold as outlined below)

AND

- meet our editorial thresholds as outlined below,

then we would be happy to look at a revised manuscript.

In the following, we list our minimum requirements for publication.

******Provide novel and fully supported insight into the lateral expansion of peatlands in high- and low-Arctic regions.**

******Outline your methodology in detail, including, statistical method, site and transect selection, justification of NDVI as an indicator, further discuss the limitation of your approach, and consider providing – ground-truthing for the early period of your analysis.**

******Ensure that your data and analysis fully support all your claims about peatland expansion; alternatively, you must tone down or remove your claims.**

If you choose to take up this option, please either highlight all changes in the manuscript text file, or provide a list of the changes to the manuscript with your responses to the reviewers.

When resubmitting, please provide a point-by-point response to the reviewers' comments. Please submit your responses as a separate file, distinct from your cover letter where you can add responses to the Editors' comments that you do not want to be made available to the reviewers. Word files are preferred.

Important: The response to reviewers must not include any figures, tables or graphs. If you wish to respond to the reviewer reports with additional data in one of these formats, please add them to the main article or Supplementary Information, and refer to them in the rebuttal. Due to current technical limitations, any figures, tables, or graphs embedded in your rebuttal will not be included in the peer review file, if published.

If the revision process takes significantly longer than three months, we will be happy to reconsider your paper at a later date, as long as nothing similar has been accepted for publication at Communications Earth & Environment or published elsewhere in the meantime.

Please use the following link to submit your revised manuscript, point-by-point response to the reviewers' comments with a list of your changes to the manuscript text (which should be in a separate document to any cover letter), a tracked-changes version of the manuscript (as a PDF file) and any completed checklist:

Link Redacted

Please do not hesitate to contact us if you have any questions or would like to discuss the required revisions further. Thank you for the opportunity to review your work.

Best regards,

Martina Grecequet, PhD
Associate Editor,
Communications Earth & Environment
@CommsEarth

EDITORIAL POLICIES AND FORMAT

If you decide to resubmit your paper, please ensure that your manuscript complies with our editorial policies and complete and upload the checklist below as a Related Manuscript file type with the revised article:

Editorial Policy Policy requirements
(Download the link to your computer as a PDF.)

- Behavioural and social science
- Ecological, evolutionary & environmental sciences
- Life sciences

<https://www.nature.com/documents/nr-reporting-summary.zip>

For your information, you can find some guidance regarding format requirements summarized on the following checklist: (<https://www.nature.com/documents/commsj-phys-style-formatting-checklist-article.pdf>) and formatting guide (<https://www.nature.com/documents/commsj-phys-style-formatting-guide-accept.pdf>).

REVIEWER COMMENTS:

Reviewer #1 (Remarks to the Author):

This is an ambitious effort to document productivity trends both within and at the margins of high latitude peatlands. The claims are not entirely novel, but because of the geographic scope and the role of peatlands in northern C storage the findings are of broader interest. Also, there has been a lack of attention given to peatland margin evolution in the modern era. I am largely comfortable with the methods and the findings, although the dataset would be strengthened immensely by multiple transects per site (though I note the logistical challenges the authors faced).

However, the presentation is deficient in several areas, and the introduction and discussion sections require substantial rewriting (see comments below).

General Comments

Introduction

Paragraphs 2-3 are too long. Paragraph 2 does not flow logically from the topic sentence through to the closing sentence.

Recommend removing much of lines 46-57 in current form, as this is a pretty generic review of Arctic ecosystem-climate change topics. Refocus paragraph around peatlands and relate everything directly to them.

Paragraph 3. I am very familiar with the use of NDVI as a proxy for shrubification or GPP increases more broadly in the Arctic and the references the author uses, but less familiar (as I suspect some readers may be) with its use in peatlands/wetlands. Please provide some peatland/wetland-specific justification for why this is a powerful metric for this ecosystem. Also, whether the peatland carbon sink is effectively increasing is not simply a function of GPP but also respiration/methanogenesis, but the authors omit this side of the equation entirely in making their case.

85-98. One of the strengths of the paper is its broad geographic scope. Given that, it would strengthen this aspect further to put the geographic scope in a broader visual context for readers not steeped in the peatland literature. For Fig. 1, rather than show just the permafrost zone, could you shade peatlands differently? This would help the reader understand the geographic representativeness of your sites. Then, a sentence in the paragraph starting line 85 establishing why the ICAAP sites were chosen.

Results

Paragraph 2 should be moved first, as this is the main message of the paper. Consider removing Paragraph 1 entirely.

Methodology

The authors explain here why they needed to use a transect-based approach to a largely remote sensing-based study. As a reader this justification may be too late.. While reading the preceding text I was wondering why, if the edges of these peatlands are delineated, could their "sampling" not have covered the entirety of each peatland and its edges? Thus, it might be worth mentioning this earlier. Then, within the methods it would be helpful to explain how these transects were chosen. With only one per site, spatial bias is a real possibility. Even with the limitations the authors faced, though, given that they had a UAV, could they not have geolocated several more "transects" at each site to align with Landsat pixels? This study doesn't rely on physical core sampling, so it's hazy why they were so spatially limited at each site.

Also, although there is a "statistical methods" section, no formal statistical tests are discussed. A portion of the text again discusses content in the plots. Plots need to stand alone with their caption, not require returning to the text for interpretation.

Discussion

The discussion in its current form requires substantial rewriting with attention to organization and flow. The subsections are a bit distracting, in particular having one called "implications" followed by "conclusions". Consider condensing into one section.

Specific Comments

Introduction

23-29. The first sentence and third sentences should be combined.

35. Eliminate first use of "often" in this sentence

37-39. First and third clauses are repetitive. Use the more specific one.

58. Remove "Studies of past changes." This is implicit in referencing those studies.

63-69. Suggest condensing into one sentence.

69. Transition to the sentence beginning with "The vegetation composition..." is awkward.

73. Remove hyphen between well and integrate

74-76. Recommend condensing and moving to immediately follow the predictions sentences. Then condense 69-73 and use to support.

77-

95. Why the two different reference periods for High and Low Arctic?

96. eliminate hyphen between "peat" and "area"

Results

140. Remove this and all subsequent statements directing the reader to the figure (lines 155, 177 have this too). Just state the finding and provide a parenthetical to the appropriate figure.

147. Fig 3C is very difficult to interpret, I believe due to inconsistency between the axes in the inset and the actual figure. The x-axis in the inset is "pixels", whereas in the actual figure there are axis labels of "% area change". If it's pixels then the figure makes partial sense, but shouldn't extend to the left of the origin. If the x-axis is % area change, then the figure doesn't make sense, because you would have to say for "Salluit" for example, "we observed -55% of pixels with a delta NDVI of -0.1", which is nonsensical. I am guessing it is supposed to be read as % of pixels, not a delta % of pixels. Also, it is unclear and not explained in the caption why there are multiple distributions for each site, you need to explain that these represent different transects (I assume).

161-162. Don't direct the reader to "compare the..." Just make the comparison.

Fig 165. Figure 4 is clean and fairly easy to understand. You need to define the x-axis within the figure caption, though. The reader shouldn't have to go to the text to ascertain what "1" and "4" mean. You might also consider something aesthetically to make the site names stand out more from the y-axis text, perhaps shading the site names, underlining, or enlarging the font.

170-176. The language here is inefficient. The total # of transects should be established early and not need to be repeated. Choose % change or # of transects, but don't report both.

178. "Variability" in what?

182. Remove "A key observation". Let your reader decide what is key.

Discussion

196. If you haven't established that your sites are a representative sample of Arctic peatlands (see earlier comment) then the evidentiary basis for the first sentence is not strong enough.

196-199. This tiny paragraph is awkward.

198. Specify the trajectory. Everything is on a trajectory of some variety.

201-203. Probably best to frame all your sentences with the early period as the reference time, not the other way around.

Also, remove “deserves consideration” and other such language. If you include it, then the reader will consider it. 203-211. You need to explain the significance of mixed pixels to your findings, not just state that they are mixed and what the mixtures are, which your readers will already know.

216-224. This paragraph needs to be completely reworked. The presentation of the different “cases” is very distracting.

222. The sentence “we know...not presently characterized” is very inadequate evidence to support this statement.

225. A simple statement of fact does not make for an adequate topic sentence.

233. This sentence is awkward. Lead with the important finding, not the caveats for it.

251-254. Remove these two sentences, they are distracting.

268. The sentence starting “However...” Doesn’t this undercut the rationale for doing your study at all?

278-end of paragraph. Rewrite substantially. Remove “Studies agree”. Remove “here again.”

288-289. Instead of “will be...” shouldn’t the authors cite evidence?

292-297. This switch to past tense is awkward.

299. Remove “interestingly”

288-307. This paragraph doesn’t use any of the evidence presented in the paper.

332. This is a sentence fragment (starting with “Specifically...”)

Reviewer #2 (Remarks to the Author):

OVERVIEW

This is a thought-provoking manuscript on the topic of the lateral expansion of peatlands in high- and low-Arctic regions. The Arctic is changing rapidly due to climatic warming, and peatlands are being deeply impacted by this. CO₂ efflux from increased respiration may be offset, in part or in whole, by increased production and litter formation, and also lateral expansion. This study therefore considers a highly topical question that is central to an important, ongoing debate. The authors use Landsat-derived NDVI and NDMI to look for changes in vegetation greenness and soil moisture, respectively, in the centres and edges of 21 peatlands across a good range of locations around the Arctic between two periods, 1985-1995 and 2005-2015 or between 1985-1995 and 2010-2020, depending on the location. I have some concerns about the manuscript, detailed below, which I believe should be addressed before it can be published.

MAJOR COMMENTS

The central goal of the paper, and indeed the title of the manuscript, concern the lateral expansion of Arctic peatlands in recent decades. However, satellite data don’t allow the conclusive detection of peat, which is below the ground surface, without a robust ground-truthing strategy. The authors have data from recent field visits to the sites, but they do not present ground-truthing data from the early period, prohibiting a definitive before-and-after comparison of soil carbon content (for example). Instead, the authors look for changes in metrics of vegetation greenness and near-surface moisture content, which may (or may not) be associated with changes in peat extent. In some places, the authors claim to show peat expansion rather more confidently than the data would seem to allow. I suggest some rephrasing with less confident, and more balanced, language would be appropriate, and to refocus on what the paper can show confidently - that the edges of Arctic peatlands appear to be getting greener, and in some – but by no means all – cases, wetter. For example, the title of the manuscript seems to indicate that peat expansion has been measured, whereas it is surmised from satellite data that are taken to be a proxy for peat extent. An adjustment to the title would seem appropriate. The abstract is also very strident about the expansion of peat: “we show that Arctic peatlands have undergone lateral expansion over the last 39 years” and should be given more nuance and transparency in this prominent part of the article. Similarly, the start of the discussion (lines 196-199) is seemingly overconfident, and would read better with a more nuanced, transparent appraisal of the findings.

In other places, the authors give details of findings that support the hypothesis of peat expansion, but do not provide equivalent details about sites and transects that show no significant trends in NDVI or NDMI between the early and late periods, or which show significant decreases. Presenting both cases in equivalent detail would be a fairer test of the hypothesis of peat expansion, and would give better balance to the paper. The results text between lines 170 and 185 is key here. We are told on line 173 that only 24 % of sites show a significant increase in edge wetting. So how many saw a significant decrease? How many showed no significant change? On line 135, we are told that three out of seven sites see reduced NDMI between the early and late periods – are these statistically significant reductions? If so, the evidence for peat expansion there becomes rather more equivocal, especially given the problems with the use of NDVI in those locations (see next paragraph).

I wonder whether NDVI is a meaningful indicator of peat extent in areas like the Lapland sites where the peat is surrounded by green vegetation such as grasses and shrubs (see line 160 – this problem is acknowledged but not accounted for in the analysis). In such sites, maybe we can only rely on NDMI? I agree that NDVI is likely to work better in the high Arctic, where the peat is surrounded by bare ground.

I strongly recommend deleting the sentence between lines 174 and 176: if the difference in means is non-significant, then we can’t say that there has been an increase. Again, as written, this passage reads as if the authors are trying too hard to push a particular finding. I suggest striving for better balance.

SPECIFIC COMMENTS

Line 26: Probably also low soil temperatures, which limit soil respiration and so peat decomposition.

Fig. 3: An interesting, if unusual, presentation, something I have never seen before. I applaud the innovation here in showing

various dimensions of change in a single panel, and the interpretation of the figure is carefully explained both in the main text and the caption.

171: Either “statistically significantly higher”, or (better) just “significantly higher”.

196-197: Seemingly overconfident given the various uncertainties.

221: These are not really drivers, just modes of change.

277: Agreed, particularly about summer moisture being maintained, given that most results for NDMI show no significant change.

299-301: Some misunderstanding here. In regions that deglaciated after the LGM, such as much of North America, Morris et al. (2018) found that peat initiation was not linked to changes in simulated net precipitation (P minus E). However, in Western Siberia, which was not glaciated during the Devensian, peat initiation was concurrent with, and likely driven by, an uptick in net precipitation at about 11,000 yr BP, which converted the existing tundra into nascent peatlands. Given that your sites are not recently deglaciated, the precipitation-driven example of Western Siberia from Morris et al. (2018) might be more relevant to the argument here, rather than the temperature-driven mechanism.

314: Typo – think this should be “losses”.

334-336: More to the point, satellite data can't tell you for sure that peatlands are expanding laterally, you can only surmise this. Suggest rewording.

With best wishes,

Paul Morris, University of Leeds, 5th June 2024.

Communications Earth & Environment is committed to improving transparency in authorship. As part of our efforts in this direction, we are now requesting that all authors identified as ‘corresponding author’ create and link their Open Researcher and Contributor Identifier (ORCID) with their account on the Manuscript Tracking System prior to acceptance. ORCID helps the scientific community achieve unambiguous attribution of all scholarly contributions. You can create and link your ORCID from the home page of the Manuscript Tracking System by clicking on ‘Modify my Springer Nature account’ and following the instructions in the link below. Please also inform all co-authors that they can add their ORCIDs to their accounts and that they must do so prior to acceptance.

Version 1:

Decision Letter:

Dear Dr Crichton,

Your manuscript titled "Satellite data indicates recent Arctic peatland expansion with warming" has now been seen by our reviewers, whose comments appear below. In light of their advice we are delighted to say that we are happy, in principle, to publish a suitably revised version in Communications Earth & Environment.

We therefore invite you to revise your paper one last time to address the remaining concerns of our reviewers. At the same time we ask that you edit your manuscript to comply with our format requirements and to maximise the accessibility and therefore the impact of your work.

EDITORIAL REQUESTS:

****Please take care to match our formatting and policy requirements. We will check revised manuscript and return manuscripts that do not comply. Such requests will lead to delays. ****

SUBMISSION INFORMATION:

OPEN ACCESS:

Communications Earth & Environment is a fully open access journal. Articles are made freely accessible on publication. For further information about article processing charges, open access funding, and advice and support from Nature Research, please visit <https://www.nature.com/commsenv/open-access>

Link Redacted

Best regards,

Martina Grecequet, PhD
Senior Editor,
Communications Earth & Environment
@CommsEarth

REVIEWERS' COMMENTS:

Reviewer #1 (Remarks to the Author):

Note: I used the track changes version to assess the manuscript, so line numbers are those in that version and might differ from the "changes accepted" version.

Introduction:

In my opinion this is the weakest part of the paper. Unfortunately it's also a very important section. I have few issues with the other sections.

I think some of the confusion with the introduction lies with the emphasis on carbon accumulation and carbon sink trajectories and the overall framing of the paper. For example, the authors note in their rebuttal and in the paper that their focus is not on carbon accumulation, yet on line 79 they advance a hypothesis regarding C sink strength and on line 85 they link (but do not cite) lateral expansion explicitly to carbon accumulation. On line 85-87 they state the goal of the paper is testing whether lateral expansion of peatlands can be detected via satellite data, yet nowhere in the preceding text are the limitations on previous detection techniques discussed. Rather, the significance of the data that might be collected is discussed at length.

I appreciate the care the authors took in responding to general comments. However, looking back at my comments from the previous draft, the authors chose to ignore several of the specific comments in this section. This is not acceptable. Comments are intended to improve your manuscript for YOUR audience, they take time to think through and write, and they are not made flippantly. The introduction is still too long for this journal's typical format and distracting to the main thrust of the paper.

Remove paragraph 2. I maintain the opinion that this paragraph is superfluous. The authors' justification for keeping it isn't adequate for that amount of extra verbiage. Parsimony and brevity, please. Looking at other papers in the most recent edition

in this journal, introduction word counts were mostly in the 500-800 range. This manuscript's is 1228. Basically, it's one really big paragraph longer than is typical.
Consider removing all of paragraph 4 except for the last sentence and placing this sentence at the start of the last paragraph.

Results

Line 140. I also like Figure 3C aesthetically, and I understand what the authors are getting at, but the axis titles are still nonsensical and cannot be published as is. Let me illustrate. Assume your area is 100 pixels total. NDVI changes follow a bimodal distribution, with increases in 50% of the pixels between t1 and t2, and vice versa decreases in the other 50%. The increase and decrease values follow a normal distribution. Your figure should then be something like in the attached.

You can have an x-axis that is "% area" and a y-axis that is Δ NDVI, but you cannot have an x-axis that is "% area change" coupled with a y-axis that is Δ NDVI. Alternatively, you could set your Y axis to just "NDVI" and keep " Δ % area".

Discussion

I appreciate the effort the authors made to place their sites in context. For a multidisciplinary journal this is very important. As such, I recommend removing lines 199- 218, moving the first half of the second paragraph first, and placing the last sentence of paragraph 1 at the end of the new paragraph 1. Most of the current paragraph one is generic site description that is NOT critical to a general reader in this section. It belongs elsewhere. The first two sentences up to "Karlebotn" are important context to your reader.

Line 235-237. The change of reference period mid-sentence is still awkward and should be made consistent.

Changes observed per region section

Recommend moving last sentence of 1st paragraph to the front, followed by

Line 289. Reword to "For example, Svalbard exhibited..."

Then finish the paragraph with a "what it means" sentence. As it stands the paragraph is more of a straight Results paragraph.

Line 342-345. Snowmelt and snowpack are not hyphenated. Move "as NDMI" into parentheses and remove commas.

Lines 349. Remove "in these formerly colder areas"

Methods

Line 510. Change "mead" to mean.

Reviewer #2 (Remarks to the Author):

The authors have dealt convincingly with the main criticisms I made on the previous version of the manuscript: i) overconfident interpretation of satellite-based evidence for peat expansion, and ii) a somewhat opaque description of some trends that didn't fit with the hypothesis of peat expansion. I believe the manuscript is now ready for publication, subject to one small but important tweak.

The addition of radiocarbon dates from basal peat at the edges of the study sites provides conclusive evidence that these ecosystems are spreading laterally (indeed, this radiocarbon evidence is rather more valuable to the argument than the remote sensing analysis). However, reading the rebuttal letter and the revised manuscript, it was not immediately clear that these dated samples are indeed basal - this only becomes clear in the supplement. The fact that the edges of these peatlands have basal dates since 1985 - not just that they "contain" recent dates, as stated in the manuscript - is crucial to this issue, and should be clarified briefly in the main text. I recommend, somewhere in the paragraph between lines 276 and 287 of the tracked changes manuscript, adding that these are basal dates. Then the reader knows that peats have initiated recently. Otherwise it can be read as if they are dates from the middle of a peat profile which initiated before 1985, which tells us little.

Lines 102-108 - a nice, early qualification, identifying the limitations of the satellite data.

Lines 166-174 - similarly, a much fairer, more transparent description of significant/non-sig trends. A valuable addition that deals with my other main criticism of the previous draft.

Response to reviewers

In response to the suggested changes, we have:

1. Fulfilled the request for validation ('ground truth') by adding new *in situ* data from peat core monoliths (^{14}C) to support the hypothesis of pan-Arctic peatland expansion, described in the main text, and presented in the supplementary material.
2. Updated the argument in the manuscript to clarify why it is not robust to add more transects
3. Added a deeper discussion on site-representativeness, as well as re-organising the discussion section
4. Rewritten parts of the manuscript in response to the review comments
5. Toned down some of the too-definitive statements as requested by reviewer 2

Detailed responses to all points are provided below.

Specific responses to reviewers comments

Reviewer #1 (Remarks to the Author):

This is an ambitious effort to document productivity trends both within and at the margins of high latitude peatlands. The claims are not entirely novel, but because of the geographic scope and the role of peatlands in northern C storage the findings are of broader interest. Also, there has been a lack of attention given to peatland margin evolution in the modern era. I am largely comfortable with the methods and the findings, although the dataset would be strengthened immensely by multiple transects per site (though I note the logistical challenges the authors faced).

Thank you, we are happy reviewer 1 agrees that the methods and findings are ambitious and appropriate. We also agree that there has been a general lack of attention to peatland margin evolution, and as far as we are aware this is the only study in which satellite data has been employed to identify changes at the Arctic peatland edges for very recent warming. In this case, we would argue that the findings are indeed novel. We also agree that there are logistical challenges in identifying true peatland edges, and we will discuss in response to later comments from R1, why we were not able to include more transects per site (in short this relates to the reviewer's own comment that identifying peatland edges is a very challenging thing to do, requiring considerable field validation which we did not have for any further sites).

However, the presentation is deficient in several areas, and the introduction and discussion sections require substantial rewriting (see comments below).

We have improved the manuscript in response to these comments, where all changes can be seen on the "tracked_changes" version.

General Comments

Introduction

Paragraphs 2-3 are too long. Paragraph 2 does not flow logically from the topic sentence through to the closing sentence. Recommend removing much of lines 46-57 in current form, as this is a pretty generic review of Arctic ecosystem-climate change topics. Refocus paragraph around peatlands and relate everything directly to them.

We have covered these pretty generic aspects of Arctic ecosystem-climate dynamics because this is a multi-disciplinary journal, and it is important to set the scene for those

readers who have less knowledge of this particular topic. We wanted to stress the complexities of this ecosystem, and point out that there are many factors involved in how climate drives ecological response in this region.

Paragraph 3. I am very familiar with the use of NDVI as a proxy for shrubification or GPP increases more broadly in the Arctic and the references the author uses, but less familiar (as I suspect some readers may be) with its use in peatlands/wetlands. Please provide some peatland/wetland-specific justification for why this is a powerful metric for this ecosystem. Also, whether the peatland carbon sink is effectively increasing is not simply a function of GPP but also respiration/methanogenesis, but the authors omit this side of the equation entirely in making their case.

We cover these aspects in both the methodological background information/supplementary (why we use NDVI to consider changes of productivity on peatlands), and very clearly in the conclusion (“*However, from satellite data alone we cannot assess soil carbon accumulation rates, so we cannot answer whether lateral expansion or increased productivity is resulting in increased carbon accumulation*”).

Throughout the manuscript we have been very careful not to conflate what is visible at the surface (greening, expansion, increased productivity) to increased carbon accumulation below the surface. We have further stressed this point at the end of the introduction in section: “*In this method, we are not equating surface changes with below-ground carbon accumulation, but we are considering the dynamics of peatland forming plant communities, that can give an indication of possible lateral (areal) changes of the peatland.*”

85-98. One of the strengths of the paper is its broad geographic scope. Given that, it would strengthen this aspect further to put the geographic scope in a broader visual context for readers not steeped in the peatland literature. For Fig. 1, rather than show just the permafrost zone, could you shade peatlands differently? This would help the reader understand the geographic representativeness of your sites. Then, a sentence in the paragraph starting line 85 establishing why the ICAAP sites were chosen.

We agree that we could provide more information on the type of peatlands that are represented by our sites, and the other types of peatlands that are present in the Arctic, to better address the representativeness of our sites. We have created a new section in the discussion “Representativeness of the study sites” specifically adding:

“*Our study encompasses a wide range of pan-Arctic peatland types across the low and high Arctic of Europe and North America, with permafrost extents in our sites ranging from sporadic (e.g., Suossjavri), to discontinuous (e.g. Karlebotn, Kevo) and continuous (e.g., Colesdalen, Peat Qilaliariak). Our permafrost-containing sites have typical permafrost peatland landforms, including high and low-centred ice-wedge polygons (e.g., Bylot Island, and Colesdalen 1) and palsas/peat plateaus (e.g., Karlebotn), but are increasingly becoming complexes with thermokarst and unfrozen areas that are reminiscent of fens and bogs. Our study considers 16 sites and 21 transects, set out on two larger low-to-high Arctic transects (Europe, and Canada) but there is now a pressing need to extend this type of analysis to new sites with the benefit of a-priori knowledge of true peat-edges. The Landsat archive represents an excellent resource for this analysis, as we demonstrate here.*”

Results

Paragraph 2 should be moved first, as this is the main message of the paper. Consider removing Paragraph 1 entirely.

We started by the site description in the results to address to some extent the comment made just prior by reviewer 1 concerning peatland types and their appearance. As we have now reorganised the discussion section, the site description is now in the discussion (rather than results) and we indeed start the results with this paragraph 2 as suggested by reviewer 1.

Methodology

The authors explain here why they needed to use a transect-based approach to a largely remote sensing-based study. As a reader this justification may be too late.. While reading the preceding text I was wondering why, if the edges of these peatlands are delineated, could their “sampling” not have covered the entirety of each peatland and its edges? Thus, it might be worth mentioning this earlier. Then, within the methods it would be helpful to explain how these transects were chosen. With only one per site, spatial bias is a real possibility. Even with the limitations the authors faced, though, given that they had a UAV, could they not have geolocated several more “transects” at each site to align with Landsat pixels? This study doesn’t rely on physical core sampling, so it’s hazy why they were so spatially limited at each site.

We cover extensively the rationale and provide lots of information regarding the transects in the supplementary material, as is the required style for this journal. We updated the text in this section to more clearly signpost readers to this supplementary material from the main text. Further we have added a note on rationale as to why we have not included more transects in the introduction section. Also, for many sites we have 2 transects, not one.

“We limit our number of transects to those identified in the field site-visits, to eliminate a source of great uncertainty in attempting to create additional peat-edge transects based only on our remotely sensed data.”

Identifying the true location of the peatland edge is certainly a non-trivial task using remotely-sensed data, even with the benefit of the drone data. To combat this, we decided to only include transects for which we had also extracted peat monoliths as part of the ICAAP project. This way, we are sure we have the edge precisely located. Undertaking the fieldwork to do this took 2 years and therefore, it is not possible to return to sites to retrieve further edge locations. In total we have included 21 transects, which we believe is a sample size large enough to defend the conclusions which we draw. In the revised version, in addressing comments from reviewer 2, we have provided some strong validation evidence from those edge-located peat monoliths to strengthen our argument, shown in supplementary material, and referenced in the main text discussion section

“As a secondary line of evidence to support our conclusion of peatland expansion, we have considered preliminary ^{14}C data from a number of peat cores that were collected on our transects as part of an upcoming study (in Prep). The methods for peat core collection and radiocarbon calibration are provided and described in Supplementary Material C. Each of these cores contain calibrated mean ^{14}C dates that fall between the start of the Landsat dataset, 1985, and the present (Figure S10). This chronological data shows that across our transects, peat, and thus soil organic carbon, has very likely been actively accumulating since 1985, with recent accumulation evident both within the peatland and at the edges (Figure S9). Dating material using ^{14}C results in an estimation of age, and we report the range of the calibrated dates in table S4. However, taken together with the satellite data showing surface greening during the same period, these two lines of evidence together provide corroboration that a likely explanation is the expansion of the peatland.”

Also, although there is a “statistical methods” section, no formal statistical tests are discussed. A portion of the text again discusses content in the plots. Plots need to stand alone with their caption, not require returning to the text for interpretation.

We have changed the title of this section to “data treatment” to better describe the content of this paragraph. We have processed a lot of data, and this obviously needed to be described. We want to point out that we have applied a formal statistical test for the significance of changes in NDVI and NDMI (Mann-Whitney U test) as shown in fig 4. We have added this information to the data treatment section. The aid in-text in interpreting fig 3 is not necessary to understand figure 3 (we have made adjustments to figure 3 in response to reviewer 1 comments).

Discussion

The discussion in its current form requires substantial rewriting with attention to organization and flow. The subsections are a bit distracting, in particular having one called “implications” followed by “conclusions”. Consider condensing into one section.

We have altered the subsections titles to directly respond to this comment, these are now:

“Representativeness of the site pan-Arctic

How measured NDVI and NDMI can indicate changes in peatland extent

Changes observed per region

Possible future of Arctic peatlands”

We understand that authors and reviewers will often have different writing styles, but we have endeavoured to adjust our text to satisfy reviewer 1 in areas where we agree it can be improved by their suggestions. In many cases as we have made significant changes to the text, a specific comment may be redundant.

Specific Comments

Introduction

23-29. The first sentence and third sentences should be combined.

35. Eliminate first use of “often” in this sentence

changed

37-39. First and third clauses are repetitive. Use the more specific one.

58. Remove “Studies of past changes.” This is implicit in referencing those studies.

63-69. Suggest condensing into one sentence.

69. Transition to the sentence beginning with “The vegetation composition...” is awkward.

73. Remove hyphen between well and integrate

74-76. Recommend condensing and moving to immediately follow the predictions sentences. Then condense 69-73 and use to support.

changed

77-

95. Why the two different reference periods for High and Low Arctic?

This point is addressed in the supplementary material. We have added a flag to point to this more clearly in the main text. The data quality for most recent years for the high Arctic is low, due to the failure of the SCL (Scan Line Corrector) in Landsat 7, as well as cloudiness, and quality flags. To address this, we used the period 2005-2015 for High Arctic, for which many more images are available.

We have added to the text here:

“(the different recent periods are due to data quality and availability, see Methods and SI B).”

96. eliminate hyphen between “peat” and “area”

“Peat-area” is the name we have assigned to this metric/quantity, as stated clearly at the end of the introduction section.

Results

140. Remove this and all subsequent statements directing the reader to the figure (lines 155, 177 have this too). Just state the finding and provide a parenthetical to the appropriate figure.

We have removed references to the figure that do not include statement of finding.

147. Fig 3C is very difficult to interpret, I believe due to inconsistency between the axes in the inset and the actual figure. The x-axis in the inset is “pixels”, whereas in the actual figure there are axis labels of “% area change”. If it’s pixels then the figure makes partial sense, but shouldn’t extend to the left of the origin. If the x-axis is % area change, then the figure doesn’t make sense, because you would have to say for “Salluit” for example, “we observed -55% of pixels with a delta NDVI of -0.1”, which is nonsensical. I am guessing it is supposed to be read as % of pixels, not a delta % of pixels. Also, it is unclear and not explained in the caption why there are multiple distributions for each site, you need to explain that these represent different transects (I assume).

We have tried to aid in understanding of this figure with the inset provided, but we agree that the axis title does not help. We have changed the axis title in the inset distribution plot to “area”, and in the change plot to “area change”. We have also adjusted axes title in the main figure, and adjusted the caption to state there are multiple sites in each region. We note that reviewer 2 liked this figure, so we had to be careful to not make such major adjustments that compromised the clarity in other ways.

161-162. Don’t direct the reader to “compare the...” Just make the comparison.

Changed the wording

Fig 165. Figure 4 is clean and fairly easy to understand. You need to define the x-axis within the figure caption, though. The reader shouldn’t have to go to the text to ascertain what “1” and “4” mean. You might also consider something aesthetically to make the site names stand out more from the y-axis text, perhaps shading the site names, underlining, or enlarging the font.

Thank you for pointing out the lack of x-axis title, it had been cropped out mistakenly. We have made the site names bold.

170-176. The language here is inefficient. The total # of transects should be established early and not need to be repeated. Choose % change or # of transects, but don’t report both.

We reported both the % and # of transects for transparency which is important in science. The number of peat-areas and number of transects was clearly stated in the introduction section. It is also extensively explained and covered in supplementary material.

178. “Variability” in what?

Variability in the mean NDVI and NDMI values. We have added this to the text

182. Remove “A key observation”. Let your reader decide what is key.

We have changed the wording.

Discussion

196. If you haven’t established that your sites are a representative sample of Arctic peatlands (see earlier comment) then the evidentiary basis for the first sentence is not strong enough.

We have added a note on site representativeness, and changed the formatting of the discussion section, including a paragraph on site-representativeness.

196-199. This tiny paragraph is awkward.

We have removed it

198. Specify the trajectory. Everything is on a trajectory of some variety.

This sentence has been removed.

201-203. Probably best to frame all your sentences with the early period as the reference time, not the other way around. Also, remove “deserves consideration” and other such language. If you include it, then the reader will consider it.

The period of time for which we are sure the edge is the edge is the recent period. So we compare what we know is there *now* to what we think was there *then*. If we were considering a reference period, it should better be the recent. But in general we have made significant changes to the discussion sections.

203-211. You need to explain the significance of mixed pixels to your findings, not just state that they are mixed and what the mixtures are, which your readers will already know.

Introducing the concept of mixed pixels is important, we do not think every reader knows what this is (again, this is a multi-disciplinary journal). We come back to this concept later in the discussion, when we talk about the “cases”. The first point of arguing for something being significant or not is to introduce what it is.

“The spatial grain of Landsat data (30x30m per pixel) deserves consideration, because each pixel will consist of a range of plant types and, especially at peat edges, of a range of land cover types. In the High Arctic, less vegetated locations’ (such as Blomstrand and Stuphallet, and Pond Inlet) edge pixels will likely comprise a mix of sparse vegetation and bare ground. In Low Arctic or more vegetated sites (Lapland especially), edge pixels will likely comprise a mix of peat forming plants, grasses with some shrubs or possibly trees. These characteristics are indicated in the transects absolute NDVI values’ variability, as discussed in the results. Despite this heterogeneity between our sites, we still see a similar pattern: peat-area greening, across transect greening, and a general maintenance of peak-summer moisture.”

216-224. This paragraph needs to be completely reworked. The presentation of the different “cases” is very distracting.

We think this is a very powerful means of describing the meaning of the increase in NDVI from the point of view of ecology and pixel size. We think this framing is essential to the argument. We have changed some of the wording however to make our message clearer.

222. The sentence “we know...not presently characterized” is very inadequate evidence to support this statement.

There is no evidence more compelling for discounting “shrubification” than a lack of shrubs... We have reworded this sentence to show more clearly our thoughts here.

“From our site visits and drone data, we know the peat edges at our sites are not presently dominated by shrubs, especially in the high Arctic where shrubs are completely absent (see SI A), so we discount shrubification as an explanation for the measured NDVI increase.”

225. A simple statement of fact does not make for an adequate topic sentence.

We have replaced the full-stop with a comma to join this to the next sentence. It is preferable to use short sentences in many cases for ease of reading, but we can link these two to satisfy reviewer 1’s point.

233. This sentence is awkward. Lead with the important finding, not the caveats for it.

We have adjusted the wording

251-254. Remove these two sentences, they are distracting.

We have moved this whole paragraph to the “Possible future of Arctic peatlands” section.

268. The sentence starting “However...” Doesn’t this undercut the rationale for doing your study at all?

The study cited is only for Lapland, and it is addressing a different time-period and different driving mechanism. It has completely different goals, hypotheses, and geographical scope to this project, being a limited study of a single region. We are looking at recent response to anthropogenic warming over 4 Arctic regions. This study helps us to validate / corroborate our work rather than being a concern about ‘undercutting’.

278-end of paragraph. Reword substantially. Remove “Studies agree”. Remove “here again.”

We prefer to retain the original grammar, which is not incorrect.

288-289. Instead of “will be...” shouldn’t the authors cite evidence?

We have discussed and signalled extensively in the text prior (throughout our manuscript) the importance of moisture for peatlands, so we considered this had already been established.

292-297. This switch to past tense is awkward.

We have changed the tense to agree with the style we have used to discuss other cited studies.

299. Remove “interestingly”

Changed

288-307. This paragraph doesn't use any of the evidence presented in the paper.

It is standard practice to frame findings by addressing important points from published work in the discussion. As we have pointed out, these are complex ecosystems and the drivers are not well understood. This section is now more clearly flagged as considering possible future change.

332. This is a sentence fragment (starting with "Specifically...")

This was to shorten the sentence length for improved readability.

Thank you for the review

Reviewer #2 (Remarks to the Author):

OVERVIEW

This is a thought-provoking manuscript on the topic of the lateral expansion of peatlands in high- and low-Arctic regions. The Arctic is changing rapidly due to climatic warming, and peatlands are being deeply impacted by this. CO₂ efflux from increased respiration may be offset, in part or in whole, by increased production and litter formation, and also lateral expansion. This study therefore considers a highly topical question that is central to an important, ongoing debate. The authors use Landsat-derived NDVI and NDMI to look for changes in vegetation greenness and soil moisture, respectively, in the centres and edges of 21 peatlands across a good range of locations around the Arctic between two periods, 1985-1995 and 2005-2015 or between 1985-1995 and 2010-2020, depending on the location. I have some concerns about the manuscript, detailed below, which I believe should be addressed before it can be published.

We thank reviewer 2 for pointing out the timely nature of our study, and that it is part of an ongoing debate about peatland response to warming.

MAJOR COMMENTS

The central goal of the paper, and indeed the title of the manuscript, concern the lateral expansion of Arctic peatlands in recent decades. However, satellite data don't allow the conclusive detection of peat, which is below the ground surface, without a robust ground-truthing strategy. The authors have data from recent field visits to the sites, but they do not present ground-truthing data from the early period, prohibiting a definitive before-and-after comparison of soil carbon content (for example). Instead, the authors look for changes in metrics of vegetation greenness and near-surface moisture content, which may (or may not) be associated with changes in peat extent. In some places, the authors claim to show peat expansion rather more confidently than the data would seem to allow. I suggest some rephrasing with less confident, and more balanced, language would be appropriate, and to refocus on what the paper can show confidently - that the edges of Arctic peatlands appear to be getting greener, and in some - but by no means all - cases, wetter. For example, the title of the manuscript seems to indicate that peat expansion has been measured, whereas it

is surmised from satellite data that are taken to be a proxy for peat extent. An adjustment to the title would seem appropriate.

We are very careful in the selection of manuscript title, "Satellite data *indicates* recent Arctic peatland expansion with warming". The use of "indicate" is, we believe, accurate to what we show and discuss in the text. To further support our argument, we have augmented our study by including information from peat cores that were collected as part of a sister study under the ICAAP project umbrella. This data provides the "ground-truthing" (we prefer to call this validation) requested by reviewer 2, and greatly strengthens the argument. This paragraph is now in the discussion section

"As a secondary line of evidence to support our conclusion of peatland expansion, we have considered preliminary 14C data from a number of peat cores that were collected on our transects as part of an upcoming study (in Prep). The methods for peat core collection and radiocarbon calibration are provided and described in Supplementary Material C. Each of these cores contain calibrated mean 14C dates that fall between the start of the Landsat dataset, 1985, and the present (Figure S10). This chronological data shows that across our transects, peat, and thus soil organic carbon, has very likely been actively accumulating since 1985, with recent accumulation evident both within the peatland and at the edges (Figure S9). Dating material using 14C results in an estimation of age, and we report the range of the calibrated dates in table S4. However, taken together with the satellite data showing surface greening during the same period, these two lines of evidence together provide corroboration that a likely explanation is the expansion of the peatland."

We have also been careful in this revised version to be clearer about what we are measuring in the satellite data.

The abstract is also very strident about the expansion of peat: "we show that Arctic peatlands have undergone lateral expansion over the last 39 years" and should be given more nuance and transparency in this prominent part of the article. Similarly, the start of the discussion (lines 196-199) is seemingly overconfident, and would read better with a more nuanced, transparent appraisal of the findings.

We have adjusted the wording here.

*"we show that Arctic peatlands have **likely** undergone lateral expansion over the last 39 years"*

The small section previously at lines 196 to 199 has now been removed. We note that in the discussion, in the final paragraph before the Conclusions, we state clearly the limitations of our study in terms of soil carbon accumulation.

"In this study, however, we cannot make definitive statements on the impact of the observed increases to productivity and inferred peatland expansion and soil carbon accumulation, because we do not consider soil respiration or the drivers of moisture changes at each site. However, we do have evidence that indicates new peats, and thus soil organic carbon, have been actively accumulating at these sites since 1985, even at the edges (SI C)."

In other places, the authors give details of findings that support the hypothesis of peat expansion, but do not provide equivalent details about sites and transects that show no significant trends in NDVI or NDMI between the early and late periods, or which show significant decreases. Presenting both cases in equivalent detail would be a fairer test of the hypothesis of peat expansion, and would give better balance to the paper. The results text between lines 170 and 185 is key here. We are told on line 173 that only 24 % of sites show

a significant increase in edge wetting. So how many saw a significant decrease? How many showed no significant change? On line 135, we are told that three out of seven sites see reduced NDMI between the early and late periods – are these statistically significant reductions? If so, the evidence for peat expansion there becomes rather more equivocal, especially given the problems with the use of NDVI in those locations (see next paragraph).

We have added this information about sites that do not show statistically significant changes or increases. Also to note here is that a reduction in peak-summer NDMI (moisture) does not necessarily mean that we think those sites could not be expanding. It depends on the mean level of moisture of the sites, but also on the seasonal moisture patterns.

“Considering the changes between the early and late period; out of 21 transects, 14 (67%) edge pixels (points 1 in Fig 4) show statistically significantly higher NDVI in the late versus the early period, compared to 11 (52%) of the furthest-in-peat pixels (points 4 in Fig 4) (with 12 (57%) for point 2, 11 (52%) for point 3). No points showed statistically significant reduction in NDVI. Out of 21 transects, 5 (24%) edge pixels show statistically significantly higher NDMI (point 1), 7 (33%) for furthest-in-peat (point 4), (6 (29%) for point 2, 8 (38%) for point 3). Only 2 points in any position on the transects showed a statistically significant reduction in NDMI (point 2 Pond Inlet East, and point 4 Blomstand 1).”

I wonder whether NDVI is a meaningful indicator of peat extent in areas like the Lapland sites where the peat is surrounded by green vegetation such as grasses and shrubs (see line 160 – this problem is acknowledged but not accounted for in the analysis). In such sites, maybe we can only rely on NDMI? I agree that NDVI is likely to work better in the high Arctic, where the peat is surrounded by bare ground.

The Lapland sites are the most complex in our dataset. We have added basal dating information here in aiding the interpretation of the results, but we do not think relying entirely on peak-summer NDMI is a perfect solution. To remain consistent our NDMI measure is for peak summer (where peak summer is indicated by peak NDVI levels). We are not convinced that peak-summer NDMI is going to be a better indicator than NDVI. However, we agree these Lapland sites pose more questions in using NDVI. This is why throughout the discussion we clearly state our concerns for these sites, and also why we cite other studies that support our argument.

“The per-site transect results for the early and late periods for NDVI and NDMI (Fig 4) show that for both the early and late periods, edges are drier than locations further into the peatland; the transects are characterised by a reduction in peak-season NDMI in all locations moving towards the peat edge (with the exception of Kevo 1). This is also generally the case for NDVI, with NDVI decreasing towards the edge, but it also depends somewhat on vegetation cover; in areas where mosses transition to grasses/shrubs (e.g. Lapland, see SI A), the edge NDVI does not vary greatly from the in-peat pixel value. This can be seen in the difference between Lapland edge and on-peat pixels values (small), and the difference between Svalbard edge and on-peat pixels values (larger)”

“In Low Arctic or more vegetated sites (Lapland especially), edge pixels will likely comprise a mix of peat forming plants, grasses with some shrubs or possibly trees. These characteristics are indicated in the transects absolute NDVI values’ variability, as discussed in the results. Despite this heterogeneity between our sites, we still see a similar pattern: peat-area greening, across transect greening, and a general maintenance of peak-summer moisture.”

“A factor likely affecting NDMI will be sub-surface conditions, such as permafrost thaw – which is especially important in palsa mires, such as the Karlebotn Lapland site. A recent

data compilation (Ruuhijärvi et al 2022), including aerial photo survey, showed that 60-70 % of palsas have thawed in Finland over the last several decades (since 1950 or so) resulting in wetter soils. Other parts of Lapland have also evidenced widespread palsa/peat plateau degradation in recent years (Borge et al., 2017; Olvmo et al., 2020; Verdonen et al., 2023), although the Karlebotn site currently appears relatively intact.

A complicating factor for the Lapland sites in general is the presence of discontinuous permafrost landscape features, including palsas; where thawing can affect both water availability and local topography, for example creating slumps and standing water bodies. Despite this, in our study there is still a recent increase in greenness (NDVI) along the full transect in the Lapland sites since the 1985-1995 period, and also on the total peatland-area (Fig 3 and 4), but less strong than for other regions (Fig 5). We cannot rule out that this may in part be due to possible encroachment of vascular plants or shrubs, as they are present at the sites. However, peatlands are persistent and resistant in this region, and recent evidence shows sustained lateral expansion over recent millennia and up to the present day (Juselius-Rajamäki et al 2023).”

I strongly recommend deleting the sentence between lines 174 and 176: if the difference in means is non-significant, then we can't say that there has been an increase. Again, as written, this passage reads as if the authors are trying too hard to push a particular finding. I suggest striving for better balance.

We agree, we have removed this.

SPECIFIC COMMENTS

Line 26: Probably also low soil temperatures, which limit soil respiration and so peat decomposition.

Agreed, added.

Fig. 3: An interesting, if unusual, presentation, something I have never seen before. I applaud the innovation here in showing various dimensions of change in a single panel, and the interpretation of the figure is carefully explained both in the main text and the caption.

Thank you. We have made a small adjustment to the inset figure x-axis titles in response to comments from reviewer 1.

171: Either “statistically significantly higher”, or (better) just “significantly higher”.

changed

196-197: Seemingly overconfident given the various uncertainties.

Paragraph removed, and discussion section re-ordered

221: These are not really drivers, just modes of change.

Correct, but they are drivers of the change in NDVI as measured by the satellite. We have removed the word “driver” as it is not necessary.

277: Agreed, particularly about summer moisture being maintained, given that most results for NDMI show no significant change.

Thank you.

299-301: Some misunderstanding here. In regions that deglaciated after the LGM, such as much of North America, Morris et al. (2018) found that peat initiation was not linked to changes in simulated net precipitation (P minus E). However, in Western Siberia, which was not glaciated during the Devensian, peat initiation was concurrent with, and likely driven by, an uptick in net precipitation at about 11,000 yr BP, which converted the existing tundra into nascent peatlands. Given that your sites are not recently deglaciated, the precipitation-driven example of Western Siberia from Morris et al. (2018) might be more relevant to the argument here, rather than the temperature-driven mechanism.

Thank you for this insight. We have added this correction to our summary of Morris et al 2018 to the relevant section, and toned down the language on future change.

“Across the northern hemisphere, peatland initiation since the last glacial maximum has been driven by warming growing seasons, rather than any increase in effective precipitation in recently deglaciated locations (e.g. North America), but for Western Siberia peat initiation may have been driven by increased net precipitation at about 11k yr BP (Morris et al., 2018). Furthermore, moisture was not found to have a control on global-scale variations in (vertical) peat accumulation, but acted rather as an “on-off” switch (Gallego-Sala et al., 2018). As the whole of the Arctic zone is warming, and will continue to warm in the near term even with emission reductions (Cai et al 2021), together with a projected increased growing season precipitation (McCrystall et al 2021), it thus seems possible that further expansion of existing peatlands may occur, in addition to new peat initiation at sites where soil moisture conditions become favourable.”

314: Typo – think this should be “losses”.

Correct, changed

334-336: More to the point, satellite data can't tell you for sure that peatlands are expanding laterally, you can only surmise this. Suggest rewording.

We have adjusted the wording, and added the findings from the basal dating of peat monoliths to support our finding.

“In this study, however, we cannot make definitive statements on the impact of the observed increases to productivity and inferred peatland expansion and soil carbon accumulation, because we do not consider soil respiration or the drivers of moisture changes at each site. However, we do have evidence that indicates new peats, and thus soil organic carbon, have been actively accumulating at these sites since 1985, even at the edges (SI C).”

With best wishes,

Thank you for the review

Paul Morris, University of Leeds, 5th June 2024.